# Long-term trends in death and dependence after ischaemic strokes: A retrospective cohort study using the South London Stroke Register (SLSR)

**Hatem A. Wafa**[1,2,3]*, **Charles D. A. Wolfe**[1,2,3], **Ajay Bhalla**[1,4], **Yanzhong Wang**[1,2,3]

**1** School of Population Health and Environmental Sciences, King's College London, London, United Kingdom, **2** National Institute for Health Research (NIHR) Biomedical Research Centre, Guy's and St Thomas' NHS Foundation Trust and King's College London, London, United Kingdom, **3** National Institute for Health Research (NIHR) Collaboration for Leadership in Applied Health Research and Care (CLAHRC) South London, London, United Kingdom, **4** Guy's and St Thomas' NHS Foundation Trust, London, United Kingdom

* hatem.a.wafa@kcl.ac.uk

## Abstract

### Background

There have been reductions in stroke mortality over recent decades, but estimates by aetiological subtypes are limited. This study estimates time trends in mortality and functional dependence by ischaemic stroke (IS) aetiological subtype over a 16-year period.

### Methods and findings

The study population was 357,308 in 2011; 50.4% were males, 56% were white, and 25% were of black ethnic backgrounds. Population-based case ascertainment of stroke was conducted, and all participants who had their first-ever IS between 2000 and 2015 were identified. Further classification was concluded according to the underlying mechanism into large-artery atherosclerosis (LAA), cardio-embolism (CE), small-vessel occlusion (SVO), other determined aetiologies (OTH), and undetermined aetiologies (UND). Temporal trends in survival rates were examined using proportional-hazards survival modelling, adjusted for demography, prestroke risk factors, case mix variables, and processes of care. We carried out additional regression analyses to explore patterns in case-fatality rates (CFRs) at 30 days and 1 year and to explore whether these trends occurred at the expense of greater functional dependence (Barthel Index [BI] < 15) among survivors. A total of 3,128 patients with first-ever ISs were registered. The median age was 70.7 years; 50.9% were males; and 66.2% were white, 25.5% were black, and 8.3% were of other ethnic groups. Between 2000–2003 and 2012–2015, the adjusted overall mortality decreased by 24% (hazard ratio [HR] per year 0.976; 95% confidence interval [CI] 0.959–0.993). Mortality reductions were equally noted in both sexes and in the white and black populations but were only significant in CE strokes (HR per year 0.972; 95% CI 0.945–0.998) and in patients aged ≥55 years (HR per year 0.975; 95% CI 0.959–0.992). CFRs within 30 days and 1 year after an IS declined by 38% (rate ratio [RR] per year 0.962; 95% CI 0.941–0.984) and 37% (RR per

**Data Availability Statement:** The raw data for this study contain both personally identifiable and confidential clinical data. The participants of the study did not consent to sharing the information

publicly, and our ethical approvals require strict information governance procedures. Requests for data access for academic use should be made to the South London Stroke Register (SLSR) team, where data will be made available subject to academic review and acceptance of a data-sharing agreement. Information can be found/requested through the following link: https://www.kcl.ac.uk/lsm/research/divisions/hscr/research/groups/stroke.

**Funding:** The National Institute for Health Research (NIHR) Collaboration for Leadership in Applied Health Research and Care South London at King's College Hospital NHS Foundation Trust, and the NIHR Biomedical Research Centre based at Guy's and St Thomas' NHS Foundation Trust and King's College London. The funder of the study had no role in study design, data collection, data analysis, data interpretation, writing of the report, or decision to publish.

**Competing interests:** The authors have declared that no competing interests exist.

**Abbreviations:** AF, atrial fibrillation; APC, annual percent change; BI, Barthel Index; CE, cardio-embolism; CFR, case-fatality rate; CI, confidence interval; CT, computed tomography; DTN, door-to-needle; GCS, Glasgow coma scale; GP, general practitioner; GWTG, Get with the Guidelines; HASU, hyperacute stroke unit; HR, hazard ratio; IS, ischaemic stroke; LAA, large-artery atherosclerosis; mRS, modified Rankin Scale; MSS, Minnesota Stroke Survey; NHS, National Health Service; NIHSS, National Institutes of Health Stroke Scale; ONS, Office for National Statistics; OTH, other determined aetiologies; OXVASC, Oxford Vascular Study; RR, rate ratio; SLSR, South London Stroke Register; SSNAP, Sentinel Stroke National Audit Programme; STROBE, Strengthening the Reporting of Observational Studies in Epidemiology; SVO, small-vessel occlusion; TIA, transient ischaemic attack; TOAST, Trial of ORG 10172 in Acute Stroke Treatment; UND, undetermined aetiologies.

year 0.963; 95% CI 0.949–0.976), respectively. Recent IS was independently associated with a 23% reduced risk of functional dependence at 3 months after onset (RR per year 0.983; 95% CI 0.968–0.998; $p = 0.002$ for trend). The study is limited by small number of events in certain subgroups (e.g., LAA), which could have led to insufficient power to detect significant trends.

## Conclusions

Both mortality and 3-month functional dependence after IS decreased by an annual average of around 2.4% and 1.7%, respectively, during 2000–2015. Such reductions were particularly evident in strokes of CE origins and in those aged ≥55 years.

Author summary

### Why was this study done?

- Stroke is one of the top causes of death and disability.

- Death after stroke has been declining recently, but patterns by subtypes, age, sex, and ethnic group are unknown and could have been masked.

### What did the researchers do and find?

- We analysed the changes in ischaemic stroke (IS) death and disability over time by different subtypes and age, sex, and ethnic groups after accounting for the corresponding changes in demography, risk factors, stroke severity, and acute management.

- Overall, death and disability rates declined by an annual average of 2.4% and 1.7%, respectively, between 2000 and 2015.

- The reductions were equally shared among sex and ethnic groups but were more evident in strokes of cardiac origin (i.e., cardio-embolic) and in patients aged ≥65 years.

### What do these findings mean?

- There were declining trends in both death and disability after IS during the past 16 years mainly derived by reductions in cardio-embolic strokes and in patients aged ≥65 years old.

- Estimates can be used by officials to plan future health policy and to monitor the effect of new public health initiatives.

## Introduction

Stroke continues to be a major public health concern affecting more than 10 million people each year around the globe [1]. It is the second most common cause of death and the third

leading cause of long-term disability. Each year, stroke accounts for a loss of approximately 1,484 disability-adjusted life years per 100,000 people [2]. Approximately one-quarter of patients die within 1 month of stroke onset [3,4], and half of the survivors are left dependent on others for everyday activities [5,6]. Around 87% of all strokes are attributed to ischaemia and require timely management in order to improve their outcomes [1,7]. Therefore, quality improvement initiatives have included providing universal access to organised stroke services; reducing time to computed tomography (CT) scan; increasing the proportion of patients receiving intravenous thrombolysis; reducing time between arrival to emergency department and receiving the treatment (door-to-needle [DTN]); and, finally, participation in quality improvement registries such as the Sentinel Stroke National Audit Programme (SSNAP) in the United Kingdom, the Get with the Guidelines (GWTG)-Stroke registry in the United States, and the Riks-Stroke registry in Sweden [8–11]. Whether overall survival among stroke patients has improved with these efforts remains inconclusive. However, data from a number of emergency medical services suggested higher rates of survival among patients with stroke because of reduced stroke severity and advances in stroke care during recent decades. Between 1990 and 2013, global stroke mortality has reportedly declined from 113 to 67 deaths per 100,000 people [1].

Survival is also known to vary between different sectors of the population. It is well-documented that older patients have greater risk of death than younger patients [12]. Also, studies from the UK and the US have shown that people of black ethnic origin have better survival advantage than their white counterparts [3,12,13]. Suggested hypotheses attributed such ethnic disparity to the differences in receipt of effective interventions or the effect of healthy migrants, because individuals willing and able to migrate are usually healthier [14]. However, studies describing how survival trends might have changed across time in each of these demographic groups are lacking.

A recent study in England has calculated the 30-day case-fatality rate (CFR) for all strokes and reported a decline from 41.8% to 26.4% in men and 44.1% to 28.5% in women between 2001 and 2010 [15]. SSNAP data indicated a reduction in CFR by approximately 50% between 1998 and 2017 [16]. Although large and nationally representative, these studies used administrative data of those with stroke and may have excluded patients treated in the community or included patients without stroke (e.g., transient ischaemic attacks [TIAs], stroke mimics). Other investigators have reported either a decline or no change in stroke death rates [3,17–23]. Again, all these studies tended to present stroke as one category (i.e., total stroke). In addition, they have limitations because of a lack of ethnic diversity, selective inclusion of patients admitted to the hospital, or restricting analyses to younger age groups.

To date, no population-based study has examined secular trends in survival after ischaemic stroke (IS) by aetiological subtype among age, sex, and ethnic groups. Studying such trends is important for monitoring purposes, which may uncover hidden patterns that can help guide health programs, public policy, and the allocation of health services and research funding. The aims of this study were, first, to use population-based data to estimate long-term secular trends in death after IS by subtype among different patient groups in South London between 2000 and 2015. The second aim of the study was to examine the changes in dependence among IS survivors over time because a decline in mortality may translate into an increase in the proportion of survivors with a functional deficit.

## Methods

### Data source

The South London Stroke Register (SLSR) is a large, prospective, population-based, registry of patients with first-ever strokes in a defined population of inner London. The design of the

registry has been previously described in detail [24–26]. Briefly, a multiple overlapping surveillance system was established in 1995 to identify all stroke patients in a study area comprising 22 electoral wards in the north of 2 London boroughs: Lambeth and Southwark. According to 2011 census data from the Office for National Statistics (ONS), the SLSR covered an area of 357,308 residents (50.4% males); 56% were white, 25% were black (14% black African, 7% black Caribbean, and 4% other black), and 18% were of other ethnic backgrounds [27]. The demographic composition of the SLSR area has changed significantly over time: during 2000–2003, people aged <65 years composed 91.1% of the population; 50.1% were females, 62.3 were white, 28.8% were black, and 9% were other. The corresponding figures during 2012–2015 were 93.4%, 54.7%, 24.7%, and 20.5%, respectively.

Patients were identified at 5 London hospitals—2 within and 3 outside (but close to) the SLSR area—in order to enhance case ascertainment. Additional community cases were notified by regular contact with all general practitioners (GPs) within and in the borders of the study area [28]. Notification sources included the accident and emergency records, hospital wards, radiology records, death certificates, coroner records, hospital stroke registries, GP computer records, hospital medical staff, GPs and practice staff, community therapists, and bereavement officers [25]. Completeness of case ascertainment was previously estimated in our population using the indirect methods of capture recapture—approximately 80% (between 75% and 88%) [26,29,30].

All data were collected prospectively by specially trained nurses, doctors, and fieldworkers who vouch for the completeness and accuracy of the data. Whenever possible, patients were assessed within 48 hours of referral to the SLSR, and data were checked against the patients' GP and medical records [26]. Stroke diagnosis follows the World Health Organisation criteria [31]. Pathological classification was based on neuroradiology (CT/MRI scans), CSF analysis, or autopsy results, which was further verified by a study clinician. Accordingly, patients were classified into cerebral infarction, primary intracerebral haemorrhage, or subarachnoid haemorrhage, whereas cases without pathological confirmation of subtype were undefined. Subtype classification of IS was carried out—using the Trial of ORG 10172 in Acute Stroke Treatment (TOAST) criteria [32]—into (1) large-artery atherosclerosis (LAA), (2) cardio-embolism (CE), (3) small-vessel occlusion (SVO), (4) other determined aetiologies (OTH), and (5) undetermined aetiologies (UND). The proportion of IS patients who received any brain scan increased from 95% in 2000–2003 to 100% in 2012–2015; MRI uptake increased from 14% to 35%.

Information collected at initial assessment included the following:

1. Demographic variables: age (calculated as the difference between the date of birth and stroke onset), sex, and self-definition of ethnic origin (1991 census question [25]; stratified into black [African, Caribbean, and other], white, and others [Asian, Pakistani, Indian, Bangladeshi, Chinese, and other])

2. Pre-existing risk factors: smoking (current versus quitter/never), alcohol intake (≥21 units/wk for men, ≥14 units/wk for women), hypertension (general practice or hospital records of systolic blood pressure > 140 mmHg or diastolic > 90 mmHg), diabetes mellitus, hypercholesterolemia (total cholesterol concentration ≥ 6 mmol/L), myocardial infarction, TIAs, and atrial fibrillation (AF) (general practice or hospital records)

3. Clinical impairment indicators: urinary incontinence, swallow test (3-oz water swallow test), Glasgow coma scale (GCS), Barthel Index (BI) within 7 days of onset, and the National Institutes of Health Stroke Scale (NIHSS)

4. Processes of care in the acute phase: hospital admission (none, stroke unit, or other medical wards), brain imaging (CT, MRI, or both), antiplatelets, anticoagulants, and thrombolysis treatment

To evaluate functional abilities at follow-ups, BI was collected by means of face-to-face visits or postal questionnaires at 3 months and 1 year after the first IS.

Informed consent and assent, when appropriate, were obtained from all participants or from a next of kin for the individuals who were too impaired to provide written consent. Ethical approval for the study was obtained from the ethics committees of Guy's and St Thomas' Hospital Trust, King's College Hospital, Queens Square, and Westminster Hospital (London).

## Statistical analysis

Analysis was planned in October 2018. We included all first-ever IS cases between January 1, 2000, and December 31, 2015, and incorporated follow-up until March 31, 2016. Survival time was calculated from date of stroke to date of death, confirmed by the ONS. Patients with no record of death were censored at March 31, 2016. Continuous variables are summarised as mean (SD) and categorical data as count (percentage). To evaluate changes in baseline characteristics over time (by quadrennial cohorts), we used the Cochran-Armitage test of trend for categorical variables and linear regression for continuous variables.

Survival curves were constructed by time cohorts, IS subtype, age, sex, and ethnic group using the Kaplan-Meier method and the log rank tests (unadjusted). Multivariate Cox proportional-hazards models were performed to assess the independent effect of time on all-cause mortality. Variables of important prognostic value, as suggested by literature review, were included into the models, and backward elimination of the least significant variables was performed. The final models included—and were therefore adjusted to—demographic variables, prior risk factors, clinical impairment indicators, and processes of care in the acute phase. Time-by-ethnicity interaction was examined to account for the possible modification of the proportional effect of time on stroke outcomes by ethnicity.

To assess whether CFRs within 30 days and 1 year after the first-ever stroke had improved over time, multivariable Poisson regression models using generalised estimation equations were constructed for overall ISs and according to initial subtype and demographic group. We used Zou's method to directly estimate relative risks instead of odds ratios by specifying a Poisson distribution and including a robust error variance in our models [33]. Our independent variable, time in 4-year periods, was included as a categorical variable, with 2000–2003 as the reference group. We multiplied the adjusted rate ratio (RR) for each period by the observed rate in the reference group to obtain adjusted rates for the study period. These rates represent the estimated risks in each period if the patient demographic characteristics were identical to that in the reference group. We also evaluated calendar year as a continuous variable to obtain adjusted changes in annual rates. The same methods were used to estimate trends in functional dependence (BI < 15) among survivors at 3 months post stroke. In addition, the modified Rankin Scale (mRS) was derived from BI [34], which was used to explore trends in dependence defined as mRS $\geq$ 3 (results are provided in Table G in S1 Appendix).

Data were complete for all covariates and outcomes, except smoking (7.9% missing); drinking (9.9%); hypertension (1.3%); diabetes (1.6%); hypercholesterolaemia (2.3%); AF (1.6%); myocardial infarction (2.3%); TIA (1.6%); hospital admission (0.7%); use of anticoagulants (28.1%), antiplatelets (27.9%), and thrombolysis (27.1%) in the acute phase; GCS (4.4%); NIHSS (16.9%); swallow test (13.3%); urinary incontinence (4.6%); BI at 7 days (13.6%); and BI at 3 months (31.9% after adjusting for death). Analysis of missing data—comparing those with complete BI at 3 months and those without—showed similar profiles with no significant

differences in almost all baseline characteristics, including demography, risk factors, medication use, and stroke severity variables (Table F in S1 Appendix). Only age and BI (7 days) varied; those whose BI record was missing at 3 months tended to be 1.8 years younger ($p = 0.002$) and less dependent on presentation ($p = 0.003$). No significant trend in BI missingness was noted over the study period ($p = 0.14$). To minimise the potentially resulting bias from missing data, multiple imputation with chained equations was applied to generate 20 datasets. Each variable with incomplete values was imputed as a binomial using all variables in the study (including date of stroke onset). Parameter estimates were finally combined using Rubin's principles [35]; these were not meaningfully different from the results of complete case analysis of nonimputed dataset.

All statistical analyses were conducted with the use of R software version 3.4.4 (Free Software Foundation). All hypothesis tests were two-sided, with a significance level of 0.05. This study is reported as per the Strengthening the Reporting of Observational Studies in Epidemiology (STROBE) guideline (S1 Checklist).

## Results

Between 2000 and 2015, a total of 4,240 patients with a first-ever stroke were registered. Of these, 3,128 (73.8%) had an IS, 715 (16.9%) had a haemorrhagic stroke, and the remaining 397 (9.3%) were unspecified. Among all IS patients, 351 (11.2%) were LAA, 815 (26.1%) CE, 791 (25.5%) SVO, 1,089 (34.8%) UND, and 82 (2.6%) OTH. Table 1 shows temporal trends in characteristics of IS patients, grouped into 4 time periods. Although there was a time trend for younger age, fewer white ethnic groups, less smoking and drinking, and less baseline stroke severity, the prevalence of cardiovascular diseases, stroke unit admission, use of CT and MRI scans, and management with thrombolytic and anticoagulant agents in the acute phase increased significantly over time. Information on medication use prior to stroke is demonstrated in Table D and Table E in S1 Appendix. There has been a decline in the prior use of antihypertensive medications from 47.6% in 2000–2003 to 37% in 2012–2015 ($p < 0.001$). Among CE patients, 357 (44.6%) had a known history of AF. Of these, 58 (17.5%) reported having used anticoagulant medicines. The use of anticoagulants in CE patients who had had AF has increased significantly over time from 13.8% in 2000–2003 to 22.4% in 2012–2015.

The median survival time for overall ISs was 5.99 years (interquartile range, 5.46–6.62). Survival rates has significantly improved over the 16-year study period (log rank test $p < 0.0001$) (Fig 1), and the risk of death declined by around 24% between 2000–2003 and 2012–2015 (hazard ratio [HR] 0.76; 95% confidence interval [CI] 0.62–0.94; $p = 0.006$) after adjusting for temporal changes in demographic characteristics, prior risk factors, clinical impairments at the outset, and processes of care. Stratified analyses showed similar reductions in both sexes and ethnic groups and all aetiological subtypes; however, these were only significant in white ethnicity (HR per year 0.978; 95% CI 0.959–0.995; $p = 0.013$), black ethnicity (HR per year 0.966; 95% CI 0.929–0.996; $p = 0.028$), CE subtype (HR per year 0.972; 95% CI 0.945–0.998; $p = 0.034$), and patients aged $\geq 65$ years (HR per year 0.975; 95% CI 0.959–0.992; $p = 0.003$) (Fig 2).

Table 2 shows time trends in 30-day and 1-year CFRs over the study period (2000–2015). The overall CFR within 30 days was 12.5% and within 1 year was 25.2% (391 and 789 of 3,128 patients, respectively). There was a significant trend toward decreased case fatality during the study period. After adjustment for temporal trends in demographic characteristics, the overall 30-day CFR decreased by 38% from 16.4% in 2000–2003 to 10.2% in 2012–2015 (adjusted RR per year 0.962; 95% CI 0.941–0.984, $p = 0.0006$). Such trends were confirmed only in the white and male populations and in those who had CE and UND strokes (Table 2). Similarly, a 37%

**Table 1. Trends in baseline characteristics in patients with first-ever IS.**

| Characteristic | Year group | | | | p-Value for trend† |
|---|---|---|---|---|---|
| | 2000–2003 | 2004–2007 | 2008–2011 | 2012–2015 | |
| | (N = 807) | (N = 1,117) | (N = 637) | (N = 567) | |
| **Demography** | | | | | |
| Age, mean (SD) | 72.31 (13.20) | 70.90 (14.27) | 69.63 (15.81) | 69.29 (15.35) | 0.0002* |
| Female | 426 (52.8) | 524 (46.9) | 325 (51.0) | 260 (45.9) | 0.06 |
| Ethnic group | | | | | |
| White | 590 (73.1) | 769 (68.8) | 393 (61.7) | 316 (55.7) | <0.0001* |
| Black | 150 (18.6) | 262 (23.5) | 188 (29.5) | 199 (35.1) | <0.0001* |
| Other/Unknown | 67 (8.3) | 86 (7.7) | 56 (8.8) | 52 (9.2) | 0.44 |
| **TOAST subtype** | | | | | |
| LAA | 67 (8.3) | 132 (11.8) | 102 (16.0) | 50 (8.8) | 0.15 |
| CE | 225 (27.9) | 275 (24.6) | 143 (22.4) | 172 (30.3) | 0.67 |
| SVO | 226 (28.0) | 266 (23.8) | 170 (26.7) | 129 (22.8) | 0.1 |
| UND | 265 (32.8) | 407 (36.4) | 215 (33.8) | 202 (35.6) | 0.51 |
| OTH | 24 (3.0) | 37 (3.3) | 7 (1.1) | 14 (2.5) | 0.13 |
| **Pre-existing risk factors** | | | | | |
| Current drinker | 395 (52.2) | 585 (58.6) | 255 (45.9) | 225 (44.4) | <0.0001* |
| Smoker | 223 (29.9) | 302 (29.2) | 170 (29.2) | 120 (23.1) | 0.017* |
| Hypertension | 483 (60.8) | 757 (68.1) | 415 (66.6) | 397 (71.0) | 0.0004* |
| Diabetes mellitus | 154 (19.6) | 249 (22.5) | 151 (24.0) | 143 (25.6) | 0.006* |
| Hypercholesterolaemia | 115 (14.9) | 298 (26.8) | 203 (33.0) | 265 (47.7) | <0.0001* |
| AF | 144 (18.1) | 170 (15.3) | 96 (15.5) | 136 (24.5) | 0.009* |
| Myocardial infarction | 94 (11.8) | 110 (9.9) | 42 (6.9) | 60 (11.2) | 0.21 |
| TIA | 97 (12.2) | 117 (10.5) | 60 (9.7) | 57 (10.3) | 0.2 |
| **Stroke severity** | | | | | |
| Urinary incontinence | 300 (40.8) | 439 (39.8) | 184 (30.4) | 156 (28.9) | <0.0001* |
| Swallow test (fail) | 278 (38.0) | 311 (31.8) | 130 (24.5) | 99 (21.1) | <0.0001* |
| GCS < 13 | 169 (22.1) | 230 (21.1) | 122 (20.4) | 105 (19.4) | 0.22 |
| BI < 15 (7 d) | 362 (53.5) | 523 (53.6) | 237 (43.2) | 186 (37.3) | <0.0001* |
| NIHSS severity | | | | | |
| Mild (≤4) | 175 (33.7) | 415 (39.0) | 232 (41.6) | 217 (47.5) | <0.0001* |
| Moderate (5–20) | 312 (60.0) | 526 (49.4) | 268 (48.0) | 213 (46.6) | <0.0001* |
| Severe (>20) | 33 (6.3) | 123 (11.6) | 58 (10.4) | 27 (5.9) | 0.53 |
| **Processes of care in the acute phase** | | | | | |
| Admission | | | | | |
| None | 125 (15.7) | 73 (6.6) | 42 (6.6) | 31 (5.5) | <0.0001* |
| Stroke unit | 438 (55.0) | 843 (76.0) | 509 (80.0) | 477 (84.6) | <0.0001* |
| Other medical wards | 234 (29.4) | 193 (17.4) | 85 (13.4) | 56 (9.9) | <0.0001* |
| Brain imaging | | | | | |
| None | 43 (5.4) | 6 (0.5) | 3 (0.5) | 0 (0.0) | <0.0001* |
| CT | 630 (79.1) | 823 (74.0) | 296 (46.5) | 339 (59.8) | <0.0001* |
| MRI | 77 (9.7) | 63 (5.7) | 9 (1.4) | 14 (2.5) | <0.0001* |
| CT and MRI | 30 (3.8) | 216 (19.4) | 244 (38.4) | 167 (29.5) | <0.0001* |
| CT or MRI | 16 (2.0) | 4 (0.4) | 84 (13.2) | 47 (8.3) | <0.0001* |
| Antiplatelet | 31 (67.4) | 914 (87.0) | 484 (79.5) | 487 (88.5) | 0.33 |
| Anticoagulant | 4 (8.7) | 172 (16.4) | 76 (12.6) | 123 (22.3) | 0.006* |

(*Continued*)

**Table 1.** (*Continued*)

| Characteristic | Year group | | | | p-Value for trend† |
|---|---|---|---|---|---|
| | **2000–2003** | **2004–2007** | **2008–2011** | **2012–2015** | |
| | **(N = 807)** | **(N = 1,117)** | **(N = 637)** | **(N = 567)** | |
| Thrombolysis | 1 (2.1) | 74 (7.0) | 105 (16.9) | 90 (16.1) | <0.0001* |

Data are count (%) unless otherwise indicated.

†*p*-Values were obtained from the Cochran-Armitage tests of trend for categorical variables and linear regression for continuous variables.

*Denotes significant trends (*p* < 0.05).

**Abbreviations:** AF, atrial fibrillation; BI, Barthel Index; CE, cardio-embolism; CI, confidence interval; CT, computed tomography; GCS, Glasgow coma scale; IS, ischaemic stroke; LAA, large-artery atherosclerosis; NIHSS, National Institutes of Health Stroke Scale; OTH, other determined aetiologies; SVO, small-vessel occlusion; TIA, transient ischaemic attack; TOAST, Trial of ORG 10172 in Acute Stroke Treatment; UND, undetermined aetiologies

risk reduction in the 1-year CFR was noted, from 32.6% in 2000–2003 to 20.5% in 2012–2015 (Table 2), which was significant in all groups.

Although rates of survival increased, proportions of functionally dependent patients (BI < 15) among IS survivors decreased. Fig 3 illustrates the proportion of survivors with functional dependency (BI < 15) across time at several points after IS onset. Most of the improvements in functional abilities were noted in the acute stage of (7 days) and early phase of (3 months)—but not 1 year after—IS onset. A 23% risk reduction of functional dependence at 3 months was estimated over the study period (demography adjusted rate 34.7% in 2000–2003 and 26.7% in 2012–2015; adjusted RR per year 0.983; 95% CI 0.968–0.998; *p* = 0.002 for trend) (Table 3). Reductions in disability levels were observed in all groups but did not reach statistical significance in females, LAA, SVO, or UND subtypes.

## Discussion

In this population-based analysis of patients with first-ever IS, from an urban multi-ethnic population of south London, we found that risk of death after IS decreased by 24% (annual

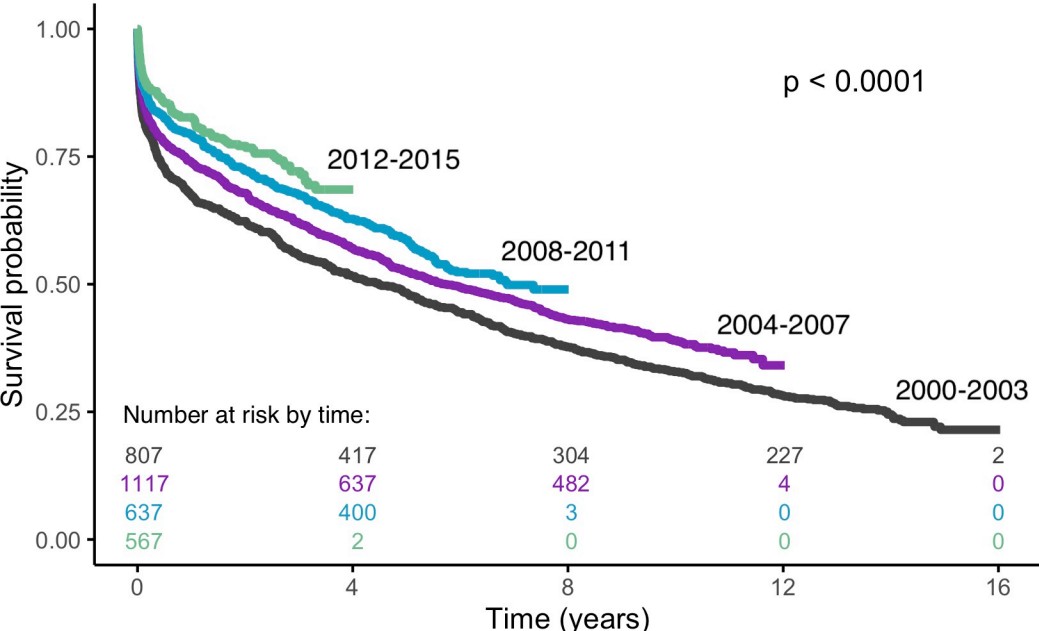

**Fig 1. Kaplan-Meier survival after IS by time cohorts.** IS, ischaemic stroke.

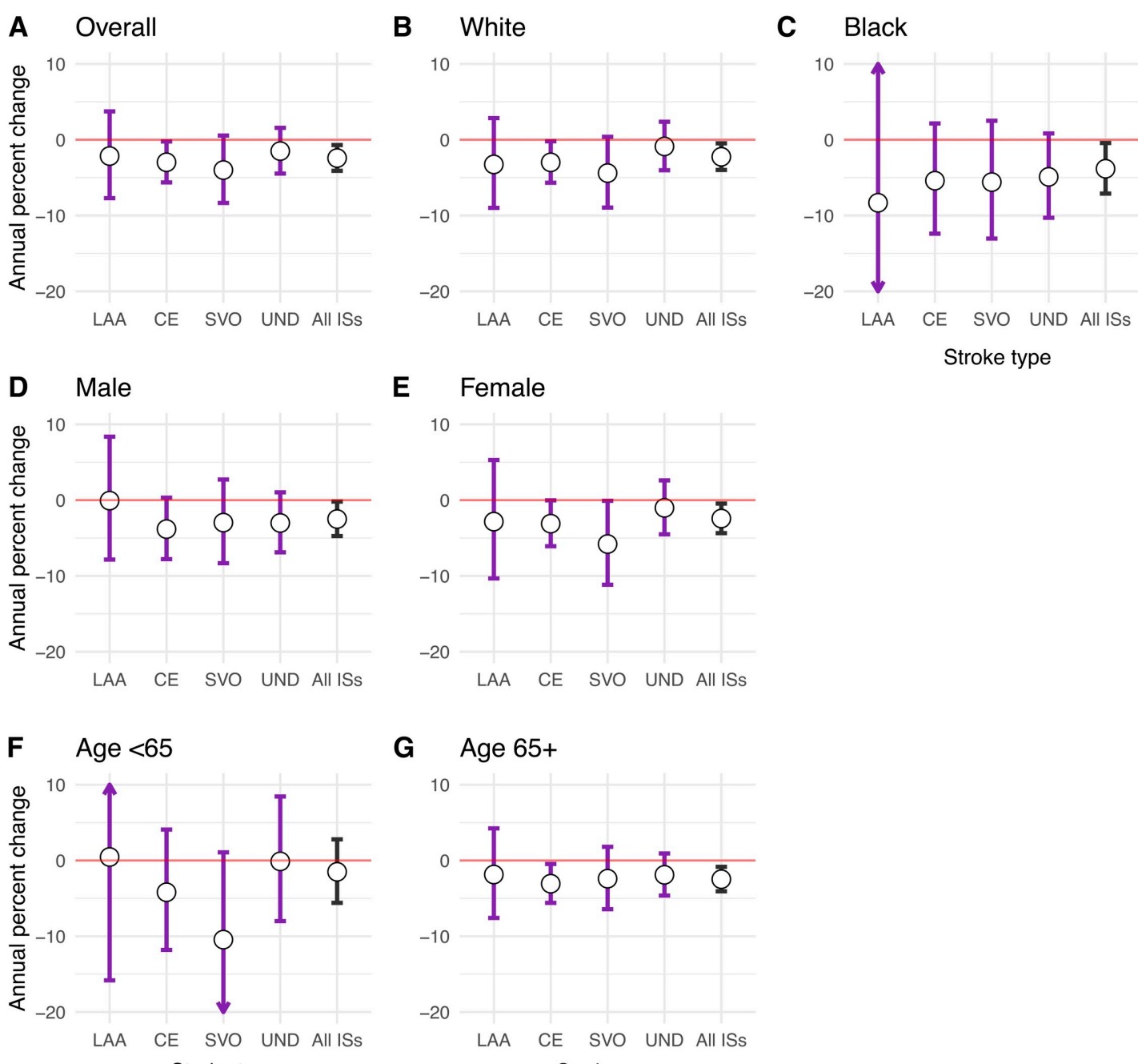

**Fig 2. Adjusted* percent change in risk of death per year among IS patients by subtype, ethnic, sex, and age groups.** Adjusted risk ratios were determined with multivariate Cox models evaluating calendar year as a continuous variable. *Adjusted for demography, prior risk factors, stroke severity, and processes of care as appropriate (see Table A, Table B, and Table C in S1 Appendix for all model covariates). Full parameter estimates are available in Table A, Table B, and Table C in S1 Appendix. CE, cardio-embolism; IS, ischaemic stroke; LAA, large-artery atherosclerosis; SVO, small-vessel occlusion; UND, undetermined aetiologies.

percent change [APC], −2.4%) between 2000 and 2015 after accounting for demography and other time-varying variables (Fig 2). Disproportionate trends were observed among aetiological subtypes for which reductions were mainly identified in strokes of CE origins (APC, −2.8%). Moreover, the rate of functional dependence at 3 months after IS declined by 23%

**Table 2. Trends in CFRs within 30 days and 1 year after the first-ever IS.**

| | Adjusted CFRs[†] | | | | Adjusted RR per year (95% CI)[‡] | p-Value for trend[§] |
|---|---|---|---|---|---|---|
| | 2000–2003 | 2004–2007 | 2008–2011 | 2012–2015 | | |
| **Within 30 d after stroke** | | | | | | |
| Overall ISs | 16.4 | 13.4 | 12.3 | 10.1 | 0.962 (0.941–0.984) | 0.0006[*] |
| By ethnicity | | | | | | |
| White | 18.3 | 15.5 | 12.7 | 11 | 0.956 (0.931–0.981) | 0.001[*] |
| Black | 6 | 6.3 | 5.3 | 4.4 | 0.977 (0.918–1.039) | 0.42 |
| By sex | | | | | | |
| Male | 12.9 | 11.5 | 10.8 | 4.7 | 0.946 (0.914–0.979) | 0.0007[*] |
| Female | 19.5 | 14.9 | 13.6 | 15.9 | 0.973 (0.945–1.002) | 0.12 |
| By age group | | | | | | |
| <55 y | 7.6 | 4.1 | 4.6 | 2.5 | 0.909 (0.843–0.98) | 0.027[*] |
| 55+ y | 19.4 | 16.6 | 14.6 | 12.9 | 0.968 (0.945–0.991) | 0.005[*] |
| By TOAST subtype | | | | | | |
| LAA[**] | 3 | 7.6 | 8.9 | 6 | 1.044 (0.953–1.143) | 0.4 |
| CE | 25.3 | 24.2 | 15.9 | 10.7 | 0.929 (0.899–0.96) | <0.0001[*] |
| SVO[**] | 1.8 | 4.6 | 4.2 | 3.2 | 1.034 (0.954–1.121) | 0.44 |
| UND | 25.7 | 13.7 | 17.2 | 15.6 | 0.963 (0.929–0.999) | 0.019[*] |
| **Within 1 y after stroke** | | | | | | |
| Overall ISs | 32.6 | 27.7 | 22.6 | 20.4 | 0.963 (0.949–0.976) | <0.0001[*] |
| By ethnicity | | | | | | |
| White | 37.1 | 30.8 | 22.7 | 24.3 | 0.959 (0.943–0.974) | <0.0001[*] |
| Black | 16.7 | 17.6 | 13.7 | 8.6 | 0.96 (0.927–0.994) | 0.009[*] |
| By sex | | | | | | |
| Male | 25.2 | 21 | 19.1 | 15.6 | 0.968 (0.946–0.989) | 0.002[*] |
| Female | 39.2 | 33.8 | 25.6 | 24.7 | 0.959 (0.942–0.976) | <0.0001[*] |
| By age group | | | | | | |
| <55 y | 11.4 | 8.4 | 7.1 | 6.2 | 0.939 (0.891–0.99) | 0.05[*] |
| 55+ y | 40 | 34.4 | 27.9 | 25.5 | 0.965 (0.952–0.979) | <0.0001[*] |
| By TOAST subtype | | | | | | |
| LAA[**] | 23.9 | 16.7 | 17.7 | 12 | 0.958 (0.904–1.015) | 0.15 |
| CE | 48 | 44.2 | 32 | 27.8 | 0.955 (0.937–0.973) | <0.0001[*] |
| SVO[**] | 11.5 | 10.1 | 10.6 | 6.2 | 0.964 (0.915–1.015) | 0.17 |
| UND | 41.1 | 31.3 | 26.4 | 21.9 | 0.956 (0.935–0.978) | <0.0001[*] |

[†]Unless otherwise indicated, figures are adjusted for demographic variables as appropriate (see Table 1). Adjusted rates for each time cohort were obtained by multiplying the observed rate for the reference period (2000–2003) by the corresponding RRs for the later periods from a model evaluating time cohorts as a categorical variable.

[‡]Adjusted risk ratios were determined with a model evaluating calendar year as a continuous variable.

[§]p-Values were obtained from the Cochran-Armitage tests for trend.

[*]Denotes significant trends ($p < 0.05$).

[**]Unadjusted because of small number of events.

**Abbreviations:** CE, cardio-embolism; CFR, case-fatality rate; CI, confidence interval; IS, ischaemic stroke; LAA, large-artery atherosclerosis; RR, rate ratio; SVO, small-vessel occlusion; TOAST, Trial of ORG 10172 in Acute Stroke Treatment; UND, undetermined aetiologies

(Table 3). Using a conservative estimate of 52,000 incident ISs per year in the UK [36], we estimate additional survival beyond 30 days and 1 year of 3,200 and 6,300 patients, respectively (based on 6.2% and 12.1% absolute improvements in the adjusted 30-day and 1-year CFRs). We also estimate aversion of more than 3,200 cases of significant disability among those who

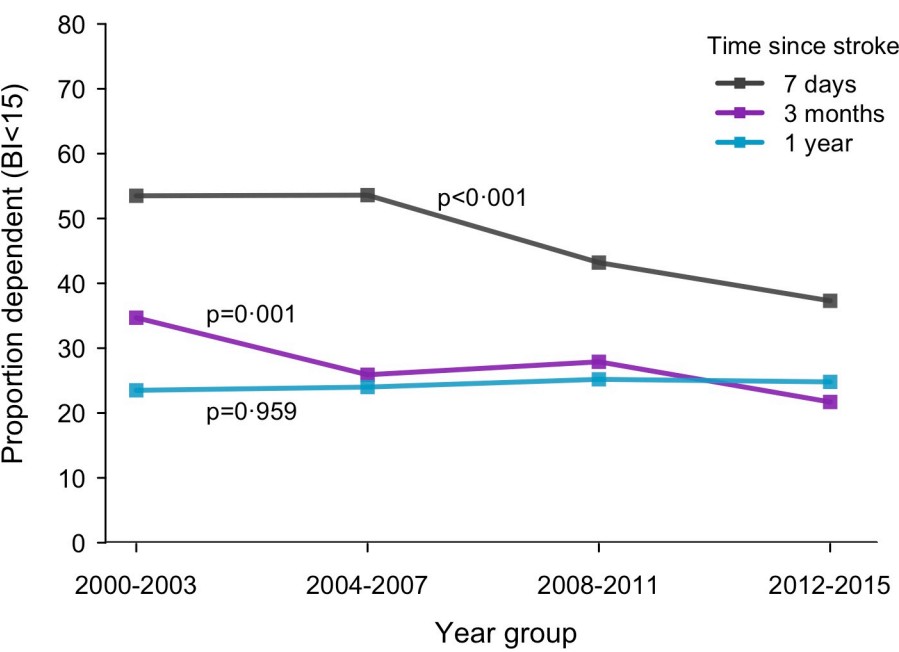

**Fig 3. Changes in the proportion of dependent (BI < 15) patients among IS survivors over time.** BI, Barthel Index; IS, ischaemic stroke.

survived up to 3 months after IS. These improved mortalities and morbidities have been accompanied by higher admission rates to hospital, increased use of CT and MRI scans, and more frequent treatment with thrombolytic and anticoagulant medications in the acute phase of stroke. However, a shift in IS characteristics toward less severe strokes was observed.

Unlike previous studies that explored trends in stroke death as a whole or by pathological types, the current analysis investigated trends in IS and its aetiological groups. The Oxford Vascular Study (OXVASC) used a modest-sized cohort of predominantly white patients and did not detect trends in early CFRs after stroke. In contrast, we estimated a significant 3.8% annual decline in the 30-day CFR after IS. Our finding corroborates national data by Seminog and colleagues suggesting a reduction of 4.7% per year in men and 4.4% per year in women (2001–2010). The steeper declines in the later study are due in large part to investigating all strokes, including haemorrhagic events, rather than conducting subtype-specific analyses. Another study in the US reported a decrease in the overall mortality after stroke by 20% for each 10-year period between 1987 and 2011 [37], which is comparable to our estimate of 24% after IS during a 16-year period. None of the previous investigations, however, were able to characterise the trends by stroke subtype due to lack of reliable diagnostic information.

Published literature from previous studies like OXVASC and the Minnesota Stroke Survey (MSS) is limited to data collected before 2005—before many current strategies for the prevention and management of stroke were implemented more widely [5,20]. For instance, national UK data from SSNAP showed an increase in thrombolysis administration rate from 1.4% in 2008 to approximately 12% in 2016 [38,39]. Similarly, our IS cohort showed increased proportion of thrombolysed patients from 2.1% in 2000–2003 to 16.1% in 2012–2015 (Table 1), approaching the estimated 20% eligibility level among acute IS cases. The treatment has a strong evidence base suggesting a 34% reduced risk of death or dependence if administered within a 3-hour time window and could therefore have contributed to the observed trends [40]. Also, the first decade of the 21st century witnessed the launch of several national

**Table 3. Trends in functional dependence (BI < 15) at initial assessment and 3 months after the first-ever IS.**

| | Adjusted disability rates (BI < 15)[†] | | | | Adjusted RR per year (95% CI)[‡] | p-Value for trend[§] |
|---|---|---|---|---|---|---|
| | 2000–2003 | 2004–2007 | 2008–2011 | 2012–2015 | | |
| **At initial assessment (7 d)** | | | | | | |
| Overall ISs | 53.5 | 52.2 | 43.7 | 39.6 | 0.976 (0.964–0.988) | <0.0001* |
| By ethnicity | | | | | | |
| White | 55.8 | 53.6 | 45.7 | 40 | 0.974 (0.959–0.989) | <0.0001* |
| Black | 47 | 47.7 | 34.6 | 36.7 | 0.977 (0.953–0.999) | 0.005* |
| By sex | | | | | | |
| Male | 46.2 | 44.4 | 33.3 | 26.1 | 0.959 (0.941–0.978) | <0.0001* |
| Female | 59.9 | 58.5 | 53.1 | 53.9 | 0.988 (0.973–0.999) | 0.043* |
| By age group | | | | | | |
| <55 y | 33.1 | 29.8 | 23.2 | 24.2 | 0.968 (0.942–0.996) | 0.013* |
| 55+ y | 60.9 | 60 | 50.8 | 44.7 | 0.977 (0.964–0.990) | <0.0001* |
| By TOAST subtype | | | | | | |
| LAA | 72.4 | 55.7 | 55.3 | 46.6 | 0.975 (0.939–0.999) | 0.009* |
| CE | 65 | 65.3 | 58.6 | 49.7 | 0.979 (0.959–0.999) | 0.0009* |
| SVO | 34.6 | 33.4 | 24.9 | 22.5 | 0.964 (0.934–0.994) | 0.004* |
| UND | 57.4 | 54.3 | 43.5 | 41 | 0.973 (0.953–0.994) | <0.0001* |
| **At 3 mo post stroke** | | | | | | |
| Overall ISs | 34.7 | 32 | 30.8 | 26.8 | 0.983 (0.968–0.998) | 0.002* |
| By ethnicity | | | | | | |
| White | 35.7 | 31.5 | 30.2 | 28.1 | 0.982 (0.964–0.999) | 0.013* |
| Black | 31.8 | 33.6 | 27.9 | 24.3 | 0.981 (0.951–0.999) | 0.043* |
| By sex | | | | | | |
| Male | 28.6 | 25.3 | 23.1 | 18.9 | 0.972 (0.948–0.996) | 0.002* |
| Female | 40.5 | 38.1 | 38.1 | 34.9 | 0.991 (0.972–1.01) | 0.17 |
| By age group | | | | | | |
| <55 y | 23.3 | 17.1 | 19.1 | 14 | 0.967 (0.932–0.999) | 0.036* |
| 55+ y | 39.3 | 37.4 | 34.8 | 31.7 | 0.985 (0.969–0.999) | 0.013* |
| By TOAST subtype | | | | | | |
| LAA** | 59 | 31.5 | 42.3 | 33.4 | 0.974 (0.929–1.021) | 0.05 |
| CE | 42 | 42.3 | 36.7 | 32.8 | 0.982 (0.958–0.999) | 0.036* |
| SVO** | 22 | 21.3 | 20 | 17.5 | 0.988 (0.94–1.017) | 0.17 |
| UND | 36.7 | 33 | 32.6 | 27.8 | 0.982 (0.958–1.007) | 0.05 |

[†]Unless otherwise indicated, figures are adjusted for demographic variables as appropriate (see Table 1). Adjusted rates for each time cohort were obtained by multiplying the observed rate for the reference period (2000–2003) by the corresponding RRs for the later periods from a model evaluating time cohorts as a categorical variable.

[‡]Adjusted risk ratios were determined with a model evaluating calendar year as a continuous variable.

[§]p-Values were obtained from the Cochran-Armitage tests for trend.

*Denotes significant trends ($p < 0.05$).

**Unadjusted because of small number of events.

**Abbreviations:** BI, Barthel Index; CE, cardio-embolism; CI, confidence interval; IS, ischaemic stroke; LAA, large-artery atherosclerosis; RR, rate ratio; SVO, small-vessel occlusion; TOAST, Trial of ORG 10172 in Acute Stroke Treatment; UND, undetermined aetiologies

awareness campaigns, including the Act FAST campaign (Face, Arms, Speech, and Time), which presumably might have had an influence on reducing time to receiving care and DTN times and subsequent mortalities [41]. Finally, stroke services in London underwent major reconfiguration in 2010 in that 8 hyperacute stroke units (HASUs)—optimally equipped and

staffed—replaced 30 hospitals used to provide acute stroke care across the city. The hospitals serving residents living in the SLSR area include 2 HASUs, while the remaining 3 hospitals have standard stroke units. This centralised model of service provision has been shown to reduce mortality [42–44].

The improved prognosis of ISs over time is likely owing to multiple factors, including not only advances in acute therapies and management but also decreased severity on presentation —genuinely driven by better primary prevention and artefactually by increased use of enhanced diagnostics methods (e.g., CT/MRI scans) enabling better detection of milder strokes [17,37,45]. Increasing emphasis has been put on measures to improve cardiovascular health at the individual and community levels [46], including smoking cessation programs and lower target levels of cholesterol and blood pressure. We have previously demonstrated increasing use of statins in our study population [36]. These agents may have beneficial effects beyond their impact on incidence and risk factors and may contribute to lower IS severity and lower subsequent morbidity and mortality [47]. Despite these efforts, countervailing trends such as the increasing prevalence of diabetes and obesity could have the opposite effect [46]. Our results show that the net effect has been a lower CFR and dependency, and that—after adjusting for the changes in demography, stroke severity, and acute phase management— there was a residual, unexplained improvement in survival after ISs during 2000–2015. Future studies are needed to better understand which specific factors have driven CFRs down so that survival gains can be consolidated and expanded.

Several issues also merit further discussion. Our results show that, for all IS subtypes, dependency levels at 3 months after the index stroke were lower in the later time cohorts than the earlier ones (Table 3). However, regression analysis showed that certain groups had steeper declines than others, which was particularly evident in CE strokes ($p$ = 0.036). The unequal declines in dependency (BI < 15) among IS subtypes may indicate differential capacity to improve and respond to interventions. CE strokes may constitute functionally disabled yet structurally intact areas the recovery of which is largely time-dependent on restoring recanalization and reperfusion. It is therefore possible that CE stroke might have benefited more from the overall advances in stroke prevention and management during recent decades [48]. However, in certain IS subtypes (e.g., LAA), power might have not been sufficient to detect significant trend because of a smaller number of events.

This population-based study describes time changes in death and dependence after IS by aetiological subtypes and demographic groups. The study was carried out in a well-defined urban area where diverse sociodemographic and economic characteristics allow generalisability of the results to other settings. Moreover, our survival trends were adjusted for—and were therefore independent of—the corresponding changes in demography, prestroke risk factors, case mix variables, and processes of care. However, our findings should be interpreted considering the following potential limitations. First, the observed trends could be in part attributable to secular changes in case ascertainment and detection rates. Increased sensitivity of surveillance methods over time would particularly improve the detection of less severe strokes. Such cases are expected to have a lower risk of death, which could have thus contributed to the observed declines in death rates. In an attempt to address the issue, we controlled for the changes in stroke severity variables. Nevertheless, certain analyses (e.g., 30-day CFR) and subgroups (e.g., LAA) had a limited number of events, and thus we were unable to account for the variations in stroke severity in these groups. Second, there are several factors that we could not adjust for in our analysis, e.g., time to hospital admission, secondary prevention, and rehabilitation received, which are also important determinants of post-IS outcomes and might explain the observed trends. Third, BI might have had suboptimal sensitivity in measuring extremes of functional deficit (ceiling and floor effects) [49]. Finally, the incomplete record of BI at 3

months may have introduced bias in the analysis of functional outcomes. In a highly dynamic population such as inner London, migration can lead to higher rates of loss to follow-up [26]. In addition, our study period covers times of great economic recession in which participants were even more reluctant to engage in research activities. However, every effort was made to keep track of all patients. Address details were updated regularly from hospital data, GP records, and through direct contact with patients or their next of kin. If patients had moved overseas, postal questionnaires were often sent and returned (S2 Appendix). No systematic loss of data was recognised; therefore, the collected information was adequate for statistical analyses. Comparing those with complete BI at 3 months and those without, both groups had similar baseline characteristics except for age and BI (7 days). Incomplete record of BI at follow-up was associated with younger age and less impaired activities of daily living on presentation. Since these variables (age and BI at 7 days) are associated with improved functional activities at follow-up, our figures regarding the prevalence of dependency among IS survivors could be an overestimate. However, our time trend estimates are unlikely to have been remarkably distorted, particularly because no significant pattern of missingness was identified over time. Furthermore, multiple imputation analysis was performed to minimise any potential bias attributed to missing data.

There is limited information on factors driving IS fatalities by aetiological subtype, including the impact of HASUs, centralised stroke care, and early rehabilitation. Exploring these would help in tailoring interventions and developing optimal care paths for each subtype of IS. In the UK, the National Health Service (NHS) has set milestones for improving stroke care over the next 10 years [50]. These include delivering a 10-fold increase in the proportion of patients who receive thrombectomy after a stroke (currently only 1% of stroke patients receive the treatment). Therefore, further reductions are expected through expanding the use of novel interventions (such as thrombectomy) and rolling out efficient models of service provision and rehabilitation. Future planning could benefit from more detailed information by IS aetiologies to develop more targeted interventions and conserve resources.

As evidenced by the results of this study, survival after IS has improved significantly in South London over the past 16 years from 2000 to 2015, but our data were not adequate to settle the issue for certain groups (e.g., LAA). Moreover, rates of functional dependence on presentation and follow-up have declined. Most of the improvements were noted in CE strokes and in patients aged ≥65 years old.

## Supporting information

**S1 STROBE Checklist. Completed STROBE checklist.** STROBE, Strengthening the Reporting of Observational Studies in Epidemiology.
(PDF)

**S1 Appendix. Supplementary tables and figures.**
(PDF)

**S2 Appendix. Postal questionnaire used for collecting information at follow-ups.**
(PDF)

## Acknowledgments

We wish to thank all the patients and their families and the healthcare professionals involved. Particular thanks to all the fieldworkers for their contributions to data collection since 1995. We would like to thank AW, MH, NW, MW, RW, and YW for support and encouragement to

start and complete the work. We also thank Hala Elwazir for enthusiasm, creative energy, and careful proofreading of the draft manuscript.

## Author Contributions

**Conceptualization:** Hatem A. Wafa, Yanzhong Wang.

**Data curation:** Hatem A. Wafa.

**Formal analysis:** Hatem A. Wafa.

**Funding acquisition:** Charles D. A. Wolfe.

**Methodology:** Hatem A. Wafa, Ajay Bhalla, Yanzhong Wang.

**Project administration:** Hatem A. Wafa, Charles D. A. Wolfe.

**Supervision:** Yanzhong Wang.

**Visualization:** Hatem A. Wafa.

**Writing – original draft:** Hatem A. Wafa.

**Writing – review & editing:** Charles D. A. Wolfe, Ajay Bhalla, Yanzhong Wang.

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
