## [Decision Letter · Decision Letter 0]

29 Nov 2019

Dear Dr. Wafa,

Thank you very much for submitting your manuscript "Long-term trends in death and dependence after ischaemic strokes: Prospective population study of the South London Stroke Register (SLSR)" (PMEDICINE-D-19-03587) for consideration at PLOS Medicine. 

[LINK]

In light of these reviews, I am afraid that we will not be able to accept the manuscript for publication in the journal in its current form, but we would like to consider a revised version that addresses the reviewers' and editors' comments. Obviously we cannot make any decision about publication until we have seen the revised manuscript and your response, and we plan to seek re-review by one or more of the reviewers. 

We expect to receive your revised manuscript by Dec 20 2019 11:59PM. Please email us (plosmedicine@plos.org) if you have any questions or concerns.

We look forward to receiving your revised manuscript. 

Sincerely,

Adya Misra, PhD

Senior Editor 

PLOS Medicine

plosmedicine.org

Title- it appears the register contains prospectively collected data but the analysis retrospective? As such the title ought to be modified to reflect this as well as include a study descriptor to adhere to PLOS Medicine style. You may consider “Long-term trends in death and dependence after ischaemic strokes: a retrospective cohort study using the South London Stroke Register (SLSR)"

Data availability statement- please provide the link for data requests and/or an accession number to identify the data set used in the study 

Author summary "At this stage, we ask that you include a short, non-technical Author Summary of your research to make findings accessible to a wide audience that includes both scientists and non-scientists. The Author Summary should immediately follow the Abstract in your revised manuscript. This text is subject to editorial change and should be distinct from the scientific abstract. Please

 see our author guidelines for more information: https://journals.plos.org/plosmedicine/s/revising-your-manuscript#loc-author-summary"

Did your study have a prospective protocol or analysis plan? Please state this (either way) early in the Methods section.

c) In either case, changes in the analysis—including those made in response to peer review comments—should be identified as such in the Methods section of the paper, with rationale.

Please ensure that the study is reported according to the STROBE guideline, and include the completed STROBE checklist as Supporting Information. When completing the checklist, please use section and paragraph numbers, rather than page numbers. Please add the following statement, or similar, to the Methods: "This study is reported as per the Strengthening the Reporting of Observational Studies in Epidemiology STROBE guideline (S1 Checklist)."

Abstract- please include brief patient demographics in the methods and findings section 

Abstract- in the methods and findings section please include the limitations of your study as the last sentence

References must be provided within square brackets and there must be a space between the text and the square bracket followed by a full stop. 

Introduction- please briefly explain door- to needle times

Methods- please clarify if any data were excluded for analysis, providing reasons 

Results- please do not use “less white patients” as an ethnic descriptor and revise as appropriate

Figure 1 requires a better explanation perhaps, specifically what are the numbers along the X-axis

Figure 3 could use X-axis labels 

Discussion- please comment on why you only studied IS trends 

Line 312 “Our results show…” 

Line 333- please avoid assertions of primacy 

Comments from the reviewers:

Reviewer #1: Long-term trends in death and dependence after ischaemic strokes: Prospective population study of the South London Stroke Register (SLSR)

This study explores death and dependence among ischemic stroke patients in South London using a population-based, registry of patient with first-ever strokes in a defined population of inner London. The aims of this study were to estimate secular trends in death after IS by subtypes among black and white patients and to examine the changes in dependence among IS survivors over time. 

1. I think the novel part about this study is the description of prognosis and mortality trends for the different subtypes of ischemic stroke. But this needs to be better explained in the background. Wy do the authors do this? Why is this important? I also lack an explanation to the aim to describe trends separately for black and white patients? This is not brought up in the background or put in a context. Do the authors expect the trends to be different? Is it socioeconomic or ethnic differences that the authors want to explore?

2. Why only IS stroke?

3. The authors write in the background when they mention previous studies on the topic that "Although large and nationally representative, these studies used administrative data and may have excluded patients treated in the community or included patients without stroke (e.g. transient ischemic attacks, stroke mimics)." 

But it is not clear to me in what way the register used in this study differ from other administrative registers and thus avoids the limitations brought up in the previous studies? For example, TIA and mimics? The authors state themselves that completeness of case ascertainment has been estimated to 80%, isn't that lower than in many administrative registers?

4. In row 67-69 the authors write "Other investigations reported conflicting results, showing either no changes or declines in death rates.5,12,15-20" But as far as I can see there are no conflicting results in these studies regarding case fatality or stroke mortality, which is what the authors refer to. All of them show improved survival after stroke over time, right? 

5. The authors report that "Between 2000 and 2015, a total of 3,843 patients with stroke was registered. Of these, 3,128 180 81.4%) had an IS, and the remaining 715 (18.6%) had haemorrhagic strokes." Were there no unclassified cases?

6. It is not completely clear if only first strokes are included, or whether there is a mix of first and recurrent? The authors sate in the methods that they included all index IS cases between Jan 1, 2000 and Dec 31, 2015. And in the discussion they say it is "first-ever" strokes. How was the procedure in order to ensure that they included first ever strokes only? This needs to be described in the manuscript. 

7. Why and when were the confounders used? Many variables are brought up in the methods section but not shown in the models. No stepwise inclusion in models are shown. Did the authors use any exclusion criteria? 

8. The authors communicate the result as if they only find improvements for CE and discuss that the unequal declines among IS subtypes may indicate differential capacity to improvement and response to interventions. However, they find declines for all strokes, even if it is statistically significant only for CE. Therefore I think it should not be over-emphasized that only CE improved, but also be discussed that the others may also have improved.

9. Seems strange to me that the mean age of stroke declined over time. The incidence of stroke has come down substantially over the past decade, which to some extent must be explained by a shift upward in the age at onset. Can this be related to how the patients were selected? I also think the mean age is quite low. Is this representative to UK overall?

10. I understand that the panels in figure 2 cannot have the exact same units at the y-axis. But if they can be made a little more similar would be helpful.

11. Discussion row 325 to 329, can the authors please expand on this? I don't understand this reasoning. 

12. Can the authors elaborate a little on the generalizability of the risk factors and the survival results? (Also related to comment 9)

13. There are result tables and figures in the discussion, which is a bit confusing to me. I think all results should be presented in the result section. And in the discussion the interpretation of them can be brought up.

14. Do the authors want to describe survival trends or explain them? A lot of the discussion and limitation section is about not being able to explain the trends, but I don´t think that is the primary aim of the paper. I think the authors should focus more on discussing comparability over the time periods such as difference in detection, that could limit comparability over the periods. That the treatments have become better is already known and one of the primary factors behind the survival improvement.

15. How to read figure 3, left panel? What does it show? And what is the Barthel index and the interpretation of it?

16. Last paragraph not really related to the paper 

17. Language editing is recommended.

Bets wishes,

Karin Modig

Associate Professor of Epidemiology, Karolinska Institutet

Reviewer #2: Based on 3128 patients with an ischemic stroke registered in the population-based South London Stroke Register (SLSR), secular trends in death after ischemic stroke by subtypes among black and white patients over a period of 16 years were estimated. Furthermore, the changes in dependence among ischemic stroke survivors over time were investigated. It was found that both mortality and functional dependence after an ischemic stroke decreased by an annual average of about 2% between 2000 and 2015. 

I have the following comments:

The Barthel index (BI) was assessed as an indicator of clinical impairment. By whom was the BI assessed at baseline examination? At follow-up, BI was collected by means of face-to-face visits or postal questionnaires. Were the information collected by a face-to-face visit comparable to the self-reported information via questionnaire? It is well known that the BI does not define sufficiently sharp classification criteria in many items, despite its widespread distribution. Thus, a course assessment by BI can be difficult.

Did the authors also assess the National Institute of Health Stroke Scale (NIHSS) and the modified rankin scale? The NIHSS is used to assess the severity, i.e. the extension of an ischemic insult. The score can be used on the one hand as a course parameter, but on the other hand also for the evaluation of a possible therapy. Furthermore, the degree of disability or dependence in daily activities is most widely used as clinical outcome measure after stroke.

The authors stratified the analyses by ethnicity and by TOAST subtypes. It would be of interest whether there are differences due to sex and age. Did the authors conduct formal tests on the interaction with sex and age? If not, this should be carried out. If there would be a significant interaction further stratified analyses should be conducted.

How was confounder selection done by the authors? 

As already mentioned by the authors, the number of events in some TOAST subgroups were very low. Thus, statements about the mortality trends in these groups were not possible.

An age-range of the study population and the proportions of males/females included in the analysis should be given in the abstract and methods section.

Reviewer #3: This is an interesting population based study on time trends in survival and outcome in subtypes of ischemic stroke. Even though the material is limited it is a well defined population and the work up of the cases seems thorough, thus the reliability on TOAST score should be good. The paper is over all well written. However, the introduction could do with some proof reading of the grammer.

In the introduction, some of the references on stroke epidemiology are rather old. A lot has changed in the last 15 years, as the present paper shows. If 15 years old references, such as 4 and 6 are used, it could be an idea to also state that they are old, particularly as the change over time is the focus of the paper.

There are a couple of issues that I fell need clarification 

1) In the methods section: for readers unfamiliar with the demographics of London, could a sentence be added about changed (or not changed) demographic in the catchment area for the study. Socioeconomic factors are important in stroke and thus this is of interest. I am a bit unclear if the data was adjusted for changed demographics over all or for the demographics of the stroke cohort. Could the authors please clarify.

2) For hypertension, was it one registered blood pressure at any time and in any situation above 140/90 mmHg that was used? 

3) In the statistics section it is stated that data was complete except for smoking (7.9%). Does that mean that there was data for 7.9% of the paritcipants or that data was missing for 7.9% of the stroke cases? Did the ompleteness of dat chage over time. Please clarify.

4) In the results: How many patients declined to participate in the study? 

5) I miss information on medication at time and before stroke onset. Was that not available? Of the CE strokes, how many had Afib known before stroke onset and were treated?In many countries, AFib is much more treated after the introduction of NOAC. Is this the case also in Southern London? Also antihypertensive medication at stroke onset would be informative. Pharmacological treament options has evolved over time and thus, the information is very relevant to the study. If not present, this is a major limitation that needs to be discussed. 

6) Do you have access to mRS? If so, it would be nice to have that included in the paper.

Except for this, I find the discussion well written and in line with the results.

Reviewer #4: I confine my remarks to statistical aspects of this paper. The general approach is fine but I have some issues to resolve before I can recommend publication.

Line 35 and many other places. Maybe give the HR (or RR) per decade rather than per year or, alternatively, give a 3rd decimal place. What's here isn't wrong, and it matches e.g. the APA style guide, but there's a big difference, over 15 years, between, e.g., 0.975 and 0.985 per year. The former gives 0.68 and the latter gives 0.80. 

Lines 47-49: Instead of (or in addition to) giving ranks of causes of death or disability, give estimates of the actual rates. Ranks have a couple problems in this context. First, *something* has to be the leading cause. Second, such rankings are very dependent on how causes are broken down. E.g. is "cancer" one cause or is "lung cancer" one cause?

Line 67-69: Did you compare effect sizes or statistical significance? If it was the latter, please reconsider. Andrew Gelman wrote an article "The Difference between 'Significant' and 'Not Significant' is not Itself Statistically Significant" https://amstat.tandfonline.com/doi/abs/10.1198/000313006X152649#.Xc1Mu9V7lPY

Line 118 and other places in that paragraph: Categorizing continuous independent variables is almost always a mistake. Frank Harrell, in *Regression Modeling Strategies* listed 11 problems with this and summed up "Nothing could be more disastrous". I wrote a blog post showing, graphically, some of the issues. https://medium.com/@peterflom/what-happens-when-we-categorize-an-independent-variable-in-regression-77d4c5862b6c

Line 149 Insert "logistic" between multivariable and regression

Line 153: Another IV that got categorized. This may be necessary for a figure or a table, but it shouldn't be used in the analysis. Use year as year (and maybe look for nonlinear effects via a spline). However, in the footnote to table 2 it says year was used as a continuous variable, so I am a little confused.

Line 160 and other places: What are these %s? Are they the amount of missing? Why was data missing? Was it MCAR, MAR, or NMAR?

Line 165-168 Significance is not really the issue here. The effect size is more important, but the missing data mechanism is critical. 

Line 203-204 Although some were stat sig and some not, the effect sizes were similar and that is more important. 

Figure 1: I'm not completely happy with this figure in general, but I'm not going to object as I don't have a better suggestion. However, please make the lines easier to distinguish. One option is to put the labels to the right of the lines, rather than on the upper right. Another would be to vary line type (e.g. use .... , ---, and so on) or use colors other than shades of blue (or, best yet, some combination)

Figure 2: I like this represenation.

Figure 3a has several problems. First, the gaps on the x-axis are wrong. The first gap (7 days to 3 months) is 1/3 as long as the second gap (3 months to 1 year) but is shown as the same size. Second, it is very hard to read. In addition to the suggestions for figure 1, you could jitter the points so that they don't completely overlap. 

Figure 3b is what's known as a "dynamite plot". They aren't a great graphical method. See http://biostat.mc.vanderbilt.edu/wiki/Main/DynamitePlots

Peter Flom

[LINK]

---

## [Decision Letter · Decision Letter 1]

20 Jan 2020

Dear Dr. Wafa,

Thank you very much for re-submitting your manuscript "Long-term trends in death and dependence after ischaemic strokes: a retrospective cohort study using the South London Stroke Register (SLSR)" (PMEDICINE-D-19-03587R1) for review by PLOS Medicine.

I have discussed the paper with my colleagues and the academic editor and it was also seen again by 3 reviewers. I am pleased to say that provided the remaining editorial and production issues are dealt with we are planning to accept the paper for publication in the journal.

[LINK]

We look forward to receiving the revised manuscript by Jan 27 2020 11:59PM. 

Sincerely,

Adya Misra, PhD

Senior Editor 

PLOS Medicine

plosmedicine.org

Requests from Editors:

Prospective data analysis plan- if no such plan exists please make sure that the Methods section transparently describes when analyses were planned, and when/why any data-driven changes to analyses took place. Changes in the analysis—including those made in response to peer

review comments—should be identified as such in the Methods section of the paper,

with rationale.

Abstract methods and findings- study area was 357,308- presumably this is the study population and not area? Please correct as needed

Abstract- please explicitly state “the limitations of this study are ….”

 Line 388 this sentence requires revision for grammar and usage “Insofar as such patients are expected to have a lower risk of death, this would contribute to the observed declines in death rates” and Line 408 “Inasmuch as these differences are associated with improved functional activities at follow up, our figures regarding…”

Line 95 - what is 'healthy migrant effect '? Please rephrase for clarity and to ensure non stigmatising labels are used

Line 131 – Please state if it is possible to disclose what the 5 hospitals are? and why 3 outside the study area and how far out - still in london? - and what was the basis for their selection / inclusion? Also line 348 states how london underwent a reconfiguration in 2010 for stroke care and 8 new units were re-equipped and replaced 30 hospitals that had previously provided stroke care. Please clarify if the 5 included in this study are part of this new structure. 

Line 403-Please ensure all questionnaires are provided as SI files or cited as a reference if previously published

Please ensure p values and 95% CI are provided throughout the main text. For example around Line 254

Comments from Reviewers:

Reviewer #2: The authors have satisfactorily answered to my comments and revised the manuscript accordingly. I have no further comments.

Reviewer #3: If it would it be possible to do an estimated mRS based on the Barthel index (for example using Wolfe et al) I Think this information could be helpful as mRS is the most used outcome measure today. Otherwise, I find my questions and concerns met

Reviewer #4: The authors have addressed my concerns and I now recommend publication

Peter Flom

[LINK]

---

## [Editor Report · Decision Letter 2]

10 Feb 2020

Dear Mr Wafa, 

On behalf of my colleagues and the academic editor, Dr. Christa Meisinger, I am delighted to inform you that your manuscript entitled "Long-term trends in death and dependence after ischaemic strokes: a retrospective cohort study using the South London Stroke Register (SLSR)" (PMEDICINE-D-19-03587R2) has been accepted for publication in PLOS Medicine. 

PRODUCTION PROCESS

PRESS

PROFILE INFORMATION

Thank you again for submitting the manuscript to PLOS Medicine. We look forward to publishing it. 

Best wishes, 

Adya Misra, PhD

Senior Editor 

PLOS Medicine

plosmedicine.org